# Distributed Unknown Input Observer-Based Global Fault-Tolerant Average Consensus Control for Linear Multi-Agent Systems

Ximing Yang
*School of Automation Engineering*
*University of Electronic Science and Technology of China*
Chengdu 611731, China
yxm961115123@163.com

Tieshan Li
*School of Automation Engineering*
*University of Electronic Science and Technology of China*
Chengdu 611731, China
tieshanli@126.com

Yue Long
*School of Automation Engineering*
*University of Electronic Science and Technology of China*
Chengdu 611731, China
longyue@uestc.edu.cn

Hanqing Yang
*School of Automation Engineering*
*University of Electronic Science and Technology of China*
Chengdu 611731, China
hqyang5517@uestc.edu.cn

*Abstract*—The paper mainly investigates the distributed unknown input observer-based global fault-tolerant average consensus control problem for multi-agent systems (MASs). First, a distributed unknown input observer based on relative estimation error is proposed, which can effectively reduce the impact of external disturbances and achieve accurate estimation of the agent states and the faults they suffered. Then, based on the obtained accurate estimations and using the relative estimation error, a global fault-tolerant average consensus controller is proposed. The proposed controller can compensate for the effects of faults and enable the MASs to achieve global average consensus. Finally, simulations are given to verify the effectiveness of the proposed scheme.

*Index Terms*—Multi-agent systems, fault-tolerant control, distributed unknown input observer, global average consensus.

## I. INTRODUCTION

In the past decades, the study of multi-agent systems (MASs) has been highly emphasized. Due to their extensive civilian and military applications, MASs are subject to stringent performance requirements, such as adaptability, flexibility, and robustness [1]. To meet these requirements, considerable attention has been given to coordination issues in MASs, such as consensus [2], containment control [3], and formation control [4]. These coordination mechanisms have been utilized in a wide range of applications such as intelligent transportation systems [5], drone formation [6], and smart grids [7]. However, the scalability and complexity of MASs render traditional centralized control schemes insufficient to meet these requirements. Therefore, the exploration

This work was supported in part by the National Natural Science Foundation of China under Grant 51939001, Grant 62273072, and Grant 62203088, in part by the Natural Science Foundation of Sichuan Province under Grant 2022NSFSC0903. (Corresponding author: Tieshan Li.)

of distributed control schemes for MASs is of significant importance.

Compared with centralized control schemes, distributed control schemes are more suitable for the coordinated control of autonomous agents in MASs [8]. Currently, the existing control schemes can be categorized into two types based on the structure of MASs: leaderless and leader-follower. The goal of control in leaderless MASs is to reach the consensus among the agents [9]. In contrast, the control objective of leader-follower MASs is for the follower agents to track the state of the leader [10]. A formation control scheme based on dynamic output feedback was proposed for cases where velocity cannot be measured, ensuring that the agents converge to the desired formation pattern within a finite time [11]. In [12], an adaptive control strategy with a fully distributed neural network was proposed to ensure that all followers track the leader's state and that the synchronization error remains within a specified range. A formation control method based on constructing a direction alignment law and formation control law using the displacement between agents was proposed to address the direction misalignment issue caused by local reference frames [13]. Overall, distributed control has emerged as a popular research direction, attracting considerable research efforts and yielding abundant results. However, many research outcomes focus solely on the control methods design and consider relatively idealized cases, assuming precise knowledge of system states and the absence of system faults, which diminishes their engineering feasibility.

In practical applications, MASs consist of numerous agents distributed across a spatial area, with each agent facing distinct environmental challenges. Agents may encounter uncertainties, such as actuator faults, which can incapacitate the entire control system [14]. To enhance the reliability and safety of the

system, it is necessary to implement measures to compensate the adverse influences of faults on the system. In this context, fault-tolerant consensus control has attracted widespread attention as an effective method to compensate for the impact of faults [15]. A virtual actuator framework-based adaptive fault-tolerant control method was proposed to achieve leader-follower consensus control under time-varying actuator faults [16]. Based on an observer framework, a reliable consensus control design method under stochastic actuator failures was proposed to achieve multi-agent consensus [17]. A distributed fault-tolerant consensus protocol based on a distributed inter-mediate observer was proposed to achieve finite-time fault-tolerant consensus control with enhanced dissipation rate [18]. However, although [18] has addressed the consensus problem of MASs under faults, they have not considered the impact of external disturbances present in practical environments on estimation performance. Fortunately, The unknown input observer, as an effective method based on disturbance decoupling technology to handle external disturbances in estimation error systems, has been widely applied [19]- [20]. Depending on [19], to address the problem of distributed secure control in MASs, a decentralized unknown input observer-based distributed secure control scheme was proposed [21].

Based on these observations, a distributed unknown input observer and a fault-tolerant average consensus controller based on relative estimation error are proposed in this paper. Major contributions of this work are summarized below:

(1) Compared with reference [18], a control scheme utilizing disturbance decoupling technology to handle external disturbances is proposed. This scheme effectively reduces the adverse influence of disturbances on estimation performance and achieves global average consensus for MASs.

(2) Distinguished from [21], a novel distributed unknown input observer utilizing relative estimation error is proposed to obtain the estimations of the state and the fault experienced by each agent. Specifically, it uses relative estimation error to determine fault estimation, incorporating output estimates rather than just the outputs themselves into the distributed algorithm.

The structure is given as follows: Section II presents the problem formulation and give some useful assumptions. In Section III, the main results including distributed unknown input observer-based global fault-tolerant average consensus control scheme and stability analysis are given. Simulations are given in Section IV. Finally, the conclusion of this work is presented in Section V.

## II. PREPARATIONS

### A. Graph Theory

An undirected graph $\mathfrak{g}$ is defined as a pair $(\upsilon, \epsilon, \mathfrak{A})$, where $\upsilon = \{\upsilon_1, \ldots, \upsilon_N\}$ represents a nonempty finite set of nodes, and $\epsilon \subseteq \upsilon \times \upsilon$ represents a set of edges. An edge $(\upsilon_i, \upsilon_j)$ denotes a pair of nodes $\upsilon_i$ and $\upsilon_j$. The adjacency matrix, denoted as $\mathfrak{A} = [a_{ij}] \in \mathbb{R}^{N \times N}$, has elements $a_{ij}$ representing the weight coefficient of the edge $(\upsilon_i, \upsilon_j)$, with $a_{ii} = 0$ and $a_{ij} = 1$ if $(\upsilon_i, \upsilon_j) \in \epsilon$. The Laplacian matrix, denoted as

$\mathfrak{L} = \mathfrak{D} - \mathfrak{A}$, is constructed where $\mathfrak{D} = [d_{ii}]$ is a diagonal matrix with $d_{ii} = \sum_{j=1}^{N} a_{ij}$.

### B. Problem Formulation

Considering a MASs with $N$ agents ($i \in \{1, \ldots, N\}$), and the dynamics of $i$th agent with actuator faults are denoted as follows:

$$\dot{x}_i(t) = Ax_i(t) + B(u_i(t) + f_i(t)) + D\omega_i(t)$$
$$y_i(t) = Cx_i(t) \tag{1}$$

where $x_i(t) \in \mathbf{R}^n$, $u_i(t) \in \mathbf{R}^m$, $y_i(t) \in \mathbf{R}^p$ represent the agent's state, input, and output, respectively. The terms $f_i(t) \in \mathbf{R}^q$ and $\omega_i(t) \in \mathbf{R}^s$ denote the actuator fault and external disturbance, respectively. The matrices $A$, $B$, $C$, and $D$ are constants with appropriate dimensions.

This paper aims to propose a global fault-tolerant average consensus controller, so that the state of all agents can achieve global average consensus, i.e., global average consensus error $\tilde{x}_i(t)$ satisfy:

$$\tilde{x}_i(t) = x_i(t) - \frac{1}{N} \sum_{i=1}^{N} x_i \Rightarrow 0. \tag{2}$$

To facilitate subsequent analysis, the following useful assumptions and lemma are given:

**Assumption 1.**

$$rank \begin{bmatrix} \mathbf{I} & D \\ C & \mathbf{0} \end{bmatrix} = n + rank(D). \tag{3}$$

**Assumption 2.** *[22] The actuator fault $f_i(t)$ is differentiable with respect to time, and its time derivative $\dot{f}_i(t)$ belongs to $L_2[0, \infty)$. Similarly, the external disturbance $\omega_i(t)$ is bounded and also belongs to $L_2[0, \infty)$.*

**Lemma 1.** *[21] For the undirected and connected graph $\mathfrak{g}$, one has $\mathfrak{L}\mathcal{M} = \mathcal{M}\mathfrak{L} = \mathfrak{L}$.*

## III. MAIN RESULTS

### A. Distributed unknown input observer-based global fault-tolerant average consensus control scheme

To reconstruct the state and actuator fault of the agent, the relative estimation error-based distributed unknown input observer for agent $i$ is proposed:

$$\dot{m}_i(t) = \Upsilon A \hat{x}_i(t) + \Upsilon B(u_i(t) + \hat{f}_i(t))$$
$$+ L_1 \left\{ \sum_{j \in N_i} a_{ij} \left[ \eta_i - \eta_j \right] \right\}$$
$$\hat{x}_i(t) = m_i(t) + \Theta y_i(t)$$
$$\dot{\hat{f}}_i(t) = L_2 \left\{ \sum_{j \in N_i} a_{ij} \left[ \eta_i - \eta_j \right] \right\}$$
$$\hat{y}_i = C \hat{x}_i(t) \tag{4}$$

where $m_i(t)$, $\hat{x}_i(t)$, $\hat{f}_i(t)$, and $\hat{y}_i$ denote the state of unknown input observer, state estimation, actuator fault estimation, and output estimation for agent $i$, respectively. And $\eta_i = y_i(t) -$

$\hat{y}_i(t)$ denotes output estimation error, $\eta_i - \eta_j$ denotes the relative estimation error. In addition, the global fault-tolerant average consensus controller for agent $i$ is proposed:

$$u_i(t) = E\hat{x}_i(t) - \hat{f}_i(t) + K\left\{ \sum_{j \in N_i} a_{ij}\left[\eta_i - \eta_j\right]\right\}. \quad (5)$$

Then, for agent $i$, the state estimation error system can be denoted as below:

$$\dot{e}_{xi}(t) = \dot{x}_i(t) - \dot{m}_i(t) - \Theta C\dot{x}_i(t). \quad (6)$$

The following condition for the matrices $\Upsilon$ and $\Theta$ can be obtained based on Assumption 1:

$$\begin{bmatrix} \Upsilon & \Theta \end{bmatrix} \begin{bmatrix} \mathbf{I} & D \\ C & \mathbf{0} \end{bmatrix} = \begin{bmatrix} \mathbf{I} & \mathbf{0} \end{bmatrix}$$

which could be re-written as follows

$$\Upsilon D = \mathbf{0}, \ \mathbf{I} - \Theta C = \Upsilon. \quad (7)$$

Then, based on the above conditions, one has

$$\begin{aligned} \dot{e}_{xi}(t) =& \Upsilon A x_i(t) + \Upsilon B(u_i(t) + f_i(t)) + \Upsilon D\omega_i(t) - \Upsilon A\hat{x}_i(t) \\ & - \Upsilon B(u_i(t) + \hat{f}_i(t)) - L_1\left\{ \sum_{j \in N_i} a_{ij}\left[\eta_i - \eta_j\right]\right\} \\ =& \Upsilon A e_{xi}(t) + \Upsilon B e_{fi}(t) \\ & - L_1 C\left\{ \sum_{j \in N_i} a_{ij}\left[e_{xi}(t) - e_{xj}(t)\right]\right\}, \end{aligned} \quad (8)$$

and the fault estimation error system can be denoted as:

$$\dot{e}_{fi}(t) = -L_2 C\left\{ \sum_{j \in N_i} a_{ij}\left[e_{xi}(t) - e_{xj}(t)\right]\right\} + \dot{f}_i(t). \quad (9)$$

Denote vector $e_i(t) = [e_{xi}^T(t), e_{fi}^T(t)]$, the augmented estimation error system can be obtained:

$$\dot{e}_i(t) = \tilde{A}e_i(t) - L\bar{C}\left\{ \sum_{j \in N_i} a_{ij}\left[e_i(t) - e_j(t)\right]\right\} + \hat{I}\dot{f}_i(t) \quad (10)$$

where

$$\tilde{A} = \begin{bmatrix} \Upsilon A & \Upsilon B \\ \mathbf{0} & \mathbf{0} \end{bmatrix}, L = \begin{bmatrix} L_1 \\ L_2 \end{bmatrix}, \bar{C} = \begin{bmatrix} C & \mathbf{0} \end{bmatrix}, \hat{I} = \begin{bmatrix} \mathbf{0} \\ \mathbf{I} \end{bmatrix}.$$

Defining vector

$$\begin{aligned} \dot{f}(t) &= \begin{bmatrix} \dot{f}_1(t) & \dots & \dot{f}_N(t) \end{bmatrix}^T, \\ e(t) &= \begin{bmatrix} e_1^T(t) & \dots & e_N^T(t) \end{bmatrix}^T. \end{aligned}$$

Then, the estimation error system can be rewritten as:

$$\dot{e}(t) = (I_N \otimes \tilde{A} - \mathfrak{L} \otimes L\bar{C})e(t) + I_N \otimes \hat{I}\dot{f}(t). \quad (11)$$

In addition, for agent $i$:, the closed-loop system can be denoted as:

$$\begin{aligned} \dot{x}_i(t) =& Ax_i(t) + B\left( E\hat{x}_i(t) - \hat{f}_i(t) + K\left\{ \sum_{j \in N_i} a_{ij}\left[\eta_i - \eta_j\right]\right\} \right. \\ & \left. + f_i(t) \right) + D\omega_i(t) \\ =& (A + BE)x_i(t) - BEe_{xi}(t) + Be_{fi}(t) \\ & + BKC\left\{ \sum_{j \in N_i} a_{ij}\left[e_{xi}(t) - e_{xj}(t)\right]\right\} + D\omega_i(t) \\ =& (A + BE)x_i(t) + \tilde{B}e_i(t) \\ & + BK\bar{C}\left\{ \sum_{j \in N_i} a_{ij}\left[e_i(t) - e_j(t)\right]\right\} + D\omega_i(t) \quad (12) \end{aligned}$$

where $\tilde{B} = [-BE \ B]$.

To achieve global average consensus, recall the global average consensus error (2) for agent $i$, defining vector

$$\begin{aligned} \tilde{x}(t) &= \begin{bmatrix} \tilde{x}_1^T(t) & \dots & \tilde{x}_N^T(t) \end{bmatrix}^T, \\ x(t) &= \begin{bmatrix} x_1^T(t) & \dots & x_N^T(t) \end{bmatrix}^T, \\ \omega(t) &= \begin{bmatrix} \omega_1^T(t) & \dots & \omega_N^T(t) \end{bmatrix}^T. \end{aligned}$$

Then, the closed-loop system can be rewritten as:

$$\begin{aligned} \dot{x}(t) =& (I_N \otimes (A + BE))x(t) + (I_N \otimes \tilde{B} \\ & + \mathfrak{L} \otimes BK\bar{C})e(t) + (I_N \otimes D)\omega(t). \quad (13) \end{aligned}$$

So, for the global average consensus error

$$\tilde{x}(t) = (\mathcal{M} \otimes I_n)x(t) \quad (14)$$

where $\mathcal{M} = I_N - \frac{1_N 1_N^T}{N}$, it can be denoted as

$$\begin{aligned} \dot{\tilde{x}}(t) =& (\mathcal{M} \otimes I_n)(I_N \otimes (A + BE))(\mathcal{M}^{-1} \otimes I_n^{-1})\tilde{x}(t) \\ & + (\mathcal{M} \otimes I_n)(I_N \otimes \tilde{B} + \mathfrak{L} \otimes BK\bar{C})e(t) \\ & + (\mathcal{M} \otimes I_n)(I_N \otimes D)\omega(t) \\ =& (I_N \otimes (A + BE))\tilde{x}(t) + (\mathcal{M} \otimes \tilde{B} + \mathfrak{L} \otimes BK\bar{C})e(t) \\ & + (\mathcal{M} \otimes D)\omega(t). \quad (15) \end{aligned}$$

### B. Stability analysis

**Theorem 1.** *For given scalar $\alpha > 0$, matrices $\Upsilon$, $\Theta$, $L$, $K$, controller feedback gain matrix $E$, Laplacian matrix $\mathfrak{L}$, matrix $\mathcal{M}$, if there exist matrices $Q = Q^T > 0$, $P = P^T > 0$ with appropriate dimensions, such that the following condition holds*

$$\Phi = \begin{bmatrix} \Phi_1 & \Phi_2 & \Phi_3 & \mathbf{0} \\ * & \Phi_4 & \mathbf{0} & \Phi_5 \\ * & * & \Phi_6 & \mathbf{0} \\ * & * & * & \Phi_7 \end{bmatrix} < 0 \quad (16)$$

*where $\Phi_1 = He\{I_N \otimes (QA + QBE)\} + \alpha I_N \otimes Q$, $\Phi_2 = \mathcal{M} \otimes Q\tilde{B} + \mathfrak{L} \otimes QBK\bar{C}$, $\Phi_3 = \mathcal{M} \otimes QD$, $\Phi_4 = He\{I_N \otimes P\tilde{A} - \mathfrak{L} \otimes PL\bar{C}\} + \alpha I_N \otimes P$, $\Phi_5 = I_N \otimes P\hat{I}$, $\Phi_6 = -I_N \otimes I_{n_\omega}$, $\Phi_7 = -I_N \otimes I_{n_f}$, then the all the signals of the estimation error*

system (11) and the global average consensus error system (15) are bounded.

*Proof.* The Lyapunov function can be chosen as below:

$$V(t) = V_1(t) + V_2(t) \tag{17}$$

where $V_1(t) = \tilde{x}^T(t)\tilde{Q}\tilde{x}(t)$, $V_2(t) = e^T(t)\tilde{P}e(t)$, $\tilde{P} = I_N \otimes P$, $\tilde{Q} = I_N \otimes Q$. Take the derivative of the above function, the following can be obtained:

$$
\begin{aligned}
\dot{V}(t) \leq & 2e^T(t)\tilde{P}\dot{e}(t) + 2\tilde{x}^T(t)\tilde{Q}\dot{\tilde{x}}(t)\\
\leq & 2e^T(t)\tilde{P}\Big((I_N \otimes \tilde{A} - \mathfrak{L} \otimes L\bar{C})e(t) + I_N \otimes \hat{I}\dot{f}(t)\Big)\\
& + 2\tilde{x}^T(t)\tilde{Q}\Big((I_N \otimes (A + BE))\tilde{x}(t)\\
& + (\mathcal{M} \otimes \tilde{B} + \mathfrak{L} \otimes BK\bar{C})e(t) + (\mathcal{M} \otimes D)\omega(t)\Big)\\
\leq & e^T(t)He\{(I_N \otimes P)(I_N \otimes \tilde{A} - \mathfrak{L} \otimes L\bar{C})\}e(t)\\
& + 2e^T(t)(I_N \otimes P)(I_N \otimes \hat{I})\dot{f}(t)\\
& + \tilde{x}^T(t)He\{(I_N \otimes Q)(I_N \otimes (A + BE))\}\tilde{x}(t)\\
& + 2\tilde{x}^T(t)(I_N \otimes Q)(\mathcal{M} \otimes \tilde{B} + \mathfrak{L} \otimes BK\bar{C})e(t)\\
& + 2\tilde{x}^T(t)(I_N \otimes Q)(\mathcal{M} \otimes D)\omega(t). \tag{18}
\end{aligned}
$$

According to the properties of the Kronecker product, we can get:

$$
\begin{aligned}
\dot{V}(t) \leq & e^T(t)He\{I_N \otimes P\tilde{A} - \mathfrak{L} \otimes PL\bar{C}\}e(t)\\
& + \tilde{x}^T(t)He\{I_N \otimes (QA + QBE)\}\tilde{x}(t)\\
& + 2\tilde{x}^T(t)(\mathcal{M} \otimes Q\tilde{B} + \mathfrak{L} \otimes QBK\bar{C})e(t)\\
& + 2\tilde{x}^T(t)(\mathcal{M} \otimes QD)\omega(t) + 2e^T(t)(I_N \otimes P\hat{I})\dot{f}(t).
\end{aligned}
$$

Define $\xi(t) = [\tilde{x}^T(t), e^T(t), \omega^T(t), \dot{f}^T(t)]$, if the following linear matrix inequality holds

$$
\Phi = \begin{bmatrix} \Phi_1 & \Phi_2 & \Phi_3 & \mathbf{0}\\ * & \Phi_4 & \mathbf{0} & \Phi_5\\ * & * & \Phi_6 & \mathbf{0}\\ * & * & * & \Phi_7 \end{bmatrix} < 0 \tag{19}
$$

where

$$
\begin{aligned}
\Phi_1 &= He\{I_N \otimes (QA + QBE)\} + \alpha I_N \otimes Q,\\
\Phi_2 &= \mathcal{M} \otimes Q\tilde{B} + \mathfrak{L} \otimes QBK\bar{C},\\
\Phi_3 &= \mathcal{M} \otimes QD,\\
\Phi_4 &= He\{I_N \otimes P\tilde{A} - \mathfrak{L} \otimes PL\bar{C}\} + \alpha I_N \otimes P,\\
\Phi_5 &= I_N \otimes P\hat{I},\\
\Phi_6 &= -I_N \otimes I_{n_\omega},\\
\Phi_7 &= -I_N \otimes I_{n_f},
\end{aligned}
$$

we have

$$
\begin{aligned}
\dot{V}(t) \leq & -\alpha e^T(t)\tilde{P}e(t) - \alpha\tilde{x}^T(t)\tilde{Q}\tilde{x}(t) + \|\omega(t)\|^2 + \|\dot{f}(t)\|^2\\
\leq & -\alpha V(t) + \Delta(t). \tag{20}
\end{aligned}
$$

As can be seen from the above conclusion, the global average consensus of MASs (1) and the boundedness of the estimation

error system (11) can be guaranteed. The proof is completed. □

Without loss of generality, the gain matrices $L$, $K$ can be solved by some algebraic operations, and the theorem is given as follows.

**Theorem 2.** *For given scalar $\alpha > 0$, matrices $\Upsilon$, $\Theta$, controller feedback gain matrix $E$, Laplacian matrix $\mathfrak{L}$, matrix $\mathcal{M}$, if there exist symmetric positive definite matrices $S$, $P$, matrices $K$, $P_L$ with appropriate dimensions, such that the following condition holds*

$$
\Psi = \begin{bmatrix} \Psi_1 & \Psi_2 & \Psi_3 & \mathbf{0}\\ * & \Psi_4 & \mathbf{0} & \Psi_5\\ * & * & \Psi_6 & \mathbf{0}\\ * & * & * & \Psi_7 \end{bmatrix} < 0 \tag{21}
$$

*where $\Psi_1 = He\{I_N \otimes (AS + BES)\} + \alpha I_N \otimes S$, $\Psi_2 = \mathcal{M} \otimes \tilde{B} + \mathfrak{L} \otimes BK\bar{C}$, $\Psi_3 = \mathcal{M} \otimes D$, $\Psi_4 = He\{I_N \otimes P\tilde{A} - \mathfrak{L} \otimes P_L\bar{C}\} + \alpha I_N \otimes P$, $\Psi_5 = I_N \otimes P\hat{I}$, $\Psi_6 = -I_N \otimes I_{n_\omega}$, $\Psi_7 = -I_N \otimes I_{n_f}$, $S = Q^{-1}$, then the all the signals of the estimation error system (11) and the global average consensus error system (15) are bounded, and gain matrix $L = P^{-1}P_L$.*

*Proof.* Post- and pre-multiplying (19) by $diag\{I_N \otimes Q^{-1}, I_N \otimes I_{n_x+n_f}, I_N \otimes I_{n_\omega}, I_N \otimes I_{n_f}\}$, the linear matrix inequality (21) can be deduced. This proof is completed. □

## IV. EXAMPLE

In this example, a group of five agents is considered. And the dynamics of the agents are in the form of

$$
\begin{aligned}
\dot{x}_i(t) =& Ax_i(t) + B(u_i(t) + f_i(t)) + D\omega_i(t)\\
y_i(t) =& Cx_i(t) \tag{22}
\end{aligned}
$$

which are borrowed from [23], and parameter matrices are given as below

$$
A = \begin{bmatrix} 0 & 1\\ 0.2 & -2 \end{bmatrix}, B = \begin{bmatrix} 0\\ 1 \end{bmatrix}, C = \begin{bmatrix} 0 & 1\\ 1 & 0 \end{bmatrix}, D = \begin{bmatrix} 0.1\\ 0.1 \end{bmatrix}.
$$

The communication graph considered in this paper is shown below:

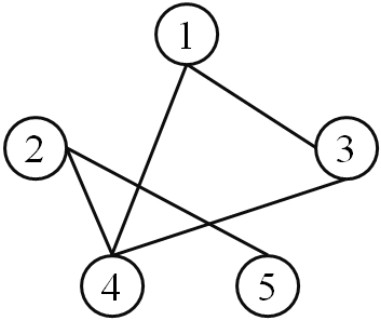

Fig. 1: Communication graph.

From Fig. 1, one has

$$\mathfrak{L} = \begin{bmatrix} 2 & 0 & -1 & -1 & 0 \\ 0 & 2 & 0 & -1 & -1 \\ -1 & 0 & 2 & -1 & 0 \\ -1 & -1 & -1 & 3 & 0 \\ 0 & -1 & 0 & 0 & 1 \end{bmatrix}.$$

To obtain the pre-design unknown input observer gain matrices, the matrix $M_\varkappa$ can be selected as follows:

$$M_\varkappa = \begin{bmatrix} -6.7245 & -9.1869 & -9.4050 & -7.5082 \\ -5.2013 & -8.2981 & -7.0737 & -8.8809 \end{bmatrix},$$

according to the following condition

$$\begin{bmatrix} \Upsilon & \Theta \end{bmatrix} = \begin{bmatrix} \mathbf{I} & \mathbf{0} \end{bmatrix} \times \begin{bmatrix} \mathbf{I} & D \\ C & \mathbf{0} \end{bmatrix}^\dagger$$
$$- M_\varkappa (\mathbf{I} - \begin{bmatrix} \mathbf{I} & D \\ C & \mathbf{0} \end{bmatrix} \times \begin{bmatrix} \mathbf{I} & D \\ C & \mathbf{0} \end{bmatrix}^\dagger),$$

the pre-design unknown input observer gain matrices can be obtained:

$$\Upsilon = \begin{bmatrix} 0.1086 & -0.1086 \\ -1.4760 & 1.4760 \end{bmatrix}, \Theta = \begin{bmatrix} 0.1086 & 0.8914 \\ -0.4760 & 1.4760 \end{bmatrix}.$$

Then, the parameters required to solve Theorem 2 are selected as $E = [-18.7279 \ -7.9363]$, $\alpha = 0.4$. the following matrices exist to make inequality (21) negative definite:

$$P = \begin{bmatrix} 22.2529 & 0.8245 & 0.2564 \\ 0.8245 & 7.9547 & -2.6069 \\ 0.2564 & -2.6069 & 1.0677 \end{bmatrix},$$

$$S = \begin{bmatrix} 14.8878 & -23.6762 \\ -23.6762 & 47.0985 \end{bmatrix},$$

$$K = \begin{bmatrix} -2.7207 & -6.7659 \end{bmatrix},$$

$$P_L = \begin{bmatrix} 0.3396 & 12.6409 \\ -0.9358 & 1.4581 \\ 6.4466 & -0.6400 \end{bmatrix}$$

where gain matrix

$$L = P^{-1}P_L = \begin{bmatrix} -0.7049 & 0.6177 \\ 9.9518 & -0.6290 \\ 30.5037 & -2.2834 \end{bmatrix}.$$

Next, experimental results are presented below to verify the effectiveness of the proposed scheme: The initial state values of agents can be selected as $x_1(0) = [8; 8]$, $x_2(0) = [8; -8]$, $x_3(0) = [-8; 8]$, $x_4(0) = [-8; -8]$, $x_5(0) = [7; 12]$. The external disturbance is $\omega_i(t) = 30\sin(2t)$, and agent 1 and 2 are considered to be faulty agents and faults they encounter are shown as follows:

$$f_1(t) = \begin{cases} 2e^{-0.1(t-5)}\sin(1.2(t-5)), & t \in [5,10] \\ 0, & otherwise \end{cases},$$

$$f_2(t) = \begin{cases} 2\sin(1.2(t-15)), & t \in [15,20] \\ 0, & otherwise \end{cases}.$$

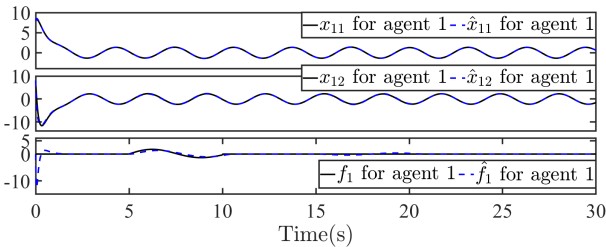

Fig. 2: Curves of state/fault and their estimations (agent 1).

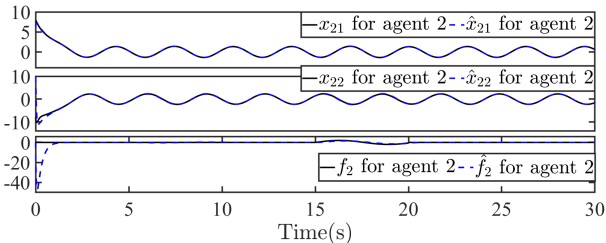

Fig. 3: Curves of state/fault and their estimations (agent 2).

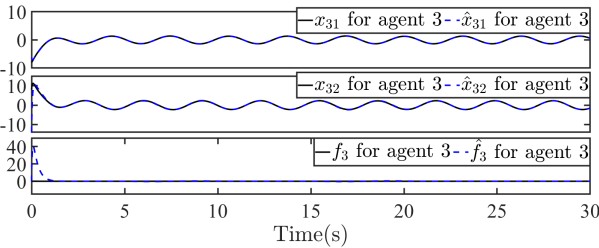

Fig. 4: Curves of state/fault and their estimations (agent 3).

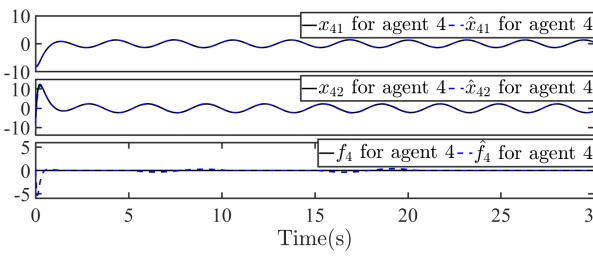

Fig. 5: Curves of state/fault and their estimations (agent 4).

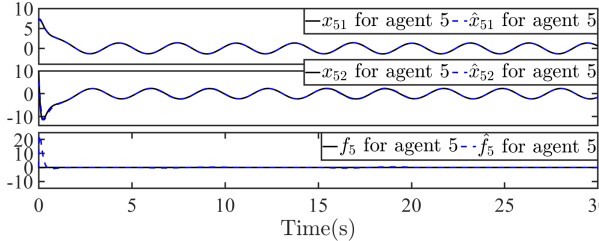

Fig. 6: Curves of state/fault and their estimations (agent 5).

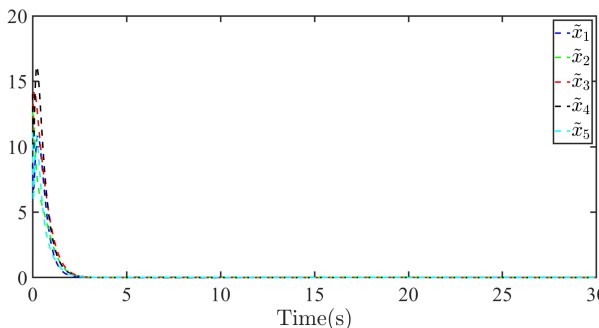

Fig. 7: Curves of global average consensus error $\tilde{x}_i(t)$.

As can be seen from Figs. 2-6, the proposed scheme (4) can effectively reduce the influence of external disturbance $\omega_i(t)$ on the estimation performance and realize accurate estimations of the agent state and fault. Based on the accurate estimations obtained by scheme (4) and the relative estimation error $\eta_i - \eta_j$, the proposed global fault-tolerant average consensus controller (5) can make the global average consensus errors $\tilde{x}_i(t)$ approach zero, as shown in Fig. 7.

## V. Conclusion

In this paper, the distributed unknown input observer-based global fault-tolerant average consensus control problem for linear MASs has been investigated. First, a distributed unknown input observer based on relative estimation error has been proposed, which can mitigate the impact of external disturbances on estimation performance, thereby achieving accurate estimations of state and fault. Then, based on the obtained estimations and the relative estimation error, a global fault-tolerant average consensus controller has been developed. The proposed scheme can compensate for fault impacts while ensuring global average consensus of the MASs. Finally, simulation experiments have been given to validate the effectiveness of the proposed control scheme.

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
