# OpenReview forum: "Distributed Unknown Input Observer-Based Global Fault-Tolerant Average Consensus Control for Linear Multi-Agent Systems"
_IEEE.org/ICIST/2024/Conference — IEEE ICIST 2024 Conference Submission_

### Official Review · Reviewer_t5AF · 2024-08-22
**Accept**

**Rating:** 7
**Confidence:** 3

**Review:**

In this paper, " Distributed Unknown Input Observer-Based Global Fault-Tolerant Average Consensus Control for Linear Multi-Agent Systems", a global fault-tolerant average consistent control method is proposed. Firstly, a distributed unknown input observer based on relative estimation error is proposed. Then, using the relative estimation error, a global fault-tolerant mean consistent controller is proposed. The article has clear logic and organization, but there are still some problems. My specific feedback is as follows :1) What are the advantages of relative estimation error compared with estimation error? 2) What are the advantages of distributed unknown input observers compared to distributed observers?

---

### Official Review · Reviewer_5tmE · 2024-08-22
**This article is quite fascinating and of high quality.**

**Rating:** 7
**Confidence:** 3

**Review:**

The paper titled "Distributed Unknown Input Observer-Based Global Fault-Tolerant Average Consensus Control for Linear Multi-Agent Systems" investigates the distributed unknown input observer-based global fault-tolerant average consensus control problem for multi-agent systems. Firstly, a distributed unknown input observer based on relative estimation errors is proposed. This observer reduces the impact of external disturbances on estimation performance, enabling accurate estimation of states and faults. A global fault-tolerant average consensus controller is designed based on the obtained estimates and relative estimation errors. Finally, the effectiveness of the proposed control scheme is validated through simulation experiments. My specific feedback is as follows: 1) Contribution Point 2 is compared with other methods, but it does not explain the advantages of the author's method. 2) What role does disturbance decoupling play in control schemes dealing with external disturbances?

---

### Official Review · Reviewer_bC3f · 2024-08-26
**Distributed Unknown Input Observer-Based Global Fault-Tolerant Average Consensus Control for Linear Multi-Agent Systems**

**Rating:** 7
**Confidence:** 2

**Review:**

The obtained result is valuable and can be accepted if the following problems can be clarified.
1. The paper should include comparisons against the existing literature to demonstrate its advantages.
2. The paper does not analyze the transient performance of the control system, which is highly influenced by the overshoots introduced by the nonlinear observer.
3. In simulation, this paper should give a result of the trajectories of leaders and followers in the multi-agent systems.

---

### Decision · Program_Chairs · 2024-09-06

Accept (Oral)